# Physiological consequences of Aldolase C deficiency during lactation

**James A. Votava[1,2], Jing Fan[2,3], Brian W. Parks[1] ***

**1** Department of Nutritional Sciences, University of Wisconsin-Madison, Madison, Wisconsin, United States of America, **2** Morgridge Institute for Research, Madison, Wisconsin, United States of America, **3** Department of Medical Microbiology and Immunology, University of Wisconsin-Madison, Madison, Wisconsin, United States of America

* brian.w.parks@wisc.edu

## Abstract

The lactating mammary gland strongly induces de novo lipogenesis (DNL) to support the synthesis of fatty acids, triglycerides, and cholesterol found within milk. In monogastric species, glucose is a major substrate utilized for DNL within the lactating mammary gland and must be efficiently taken up and processed to supply cytosolic acetyl-CoA for DNL. Along with the enzymes of the DNL pathway, the glycolytic enzyme, Aldolase C (*Aldoc*), is transcriptionally upregulated and is highly expressed during lactation in the mammary gland, suggesting a role for *Aldoc* in lactation. *Aldoc* is also a transcriptional target of the sterol regulatory element binding proteins 1 and 2 (Srebp1 and Srebp2), which transcriptionally regulate enzymes within the DNL pathway and has recently been shown to regulate plasma cholesterol and triglycerides. Here, we investigate the role of *Aldoc* in lactation, by utilizing a whole-body *Aldoc* knockout mouse. Our results demonstrate that *Aldoc* has a significant impact on lactation, whereby pups nursing from Aldoc$^{-/-}$ dams have reduced body weight. Biochemical analysis of milk identified that milk from Aldoc$^{-/-}$ dams have significantly higher galactose, lower lactose, and cholesterol content. Mass spectrometry analysis of milk lipids from Aldoc$^{-/-}$ dams revealed significantly lower quantities of medium and long chain fatty acid containing triglycerides, which has direct implications on lactation as these are the predominant triglycerides synthesized from glucose in human mammary gland. Overall, our results provide functional evidence for the contribution of *Aldoc* in mammary gland lactose and lipid synthesis during lactation.

## Introduction

The lactating mammary gland has one of the highest rates of *de novo* lipogenesis (DNL) of any tissue in the body [1, 2]. DNL is the biological process that generates fatty acids, triglycerides, and cholesterol from cytosolic acetyl-CoA. In ruminant animals, such as cows and sheep, very little carbohydrate is digested and absorbed as glucose from the diet, instead, the carbohydrate is fermented in the gut (rumen) and absorbed as short-chain fatty acids (SCFAs) [3]. These SCFAs, such as acetate, are rapidly converted to cytosolic acetyl-CoA via the enzyme Acetyl-

**Data Availability Statement:** All relevant data are within the manuscript and its Supporting Information files.

**Funding:** This work was supported in part by NIH R01-HL147097 (B.W.P).

**Competing interests:** The authors have declared that no competing interests exist.

CoA synthetase short chain family 2 (*Acss2*) and are the major substrate for mammary gland DNL in ruminants [4, 5]. However, in nonruminant, monogastric species like humans and rodents, a large amount of dietary carbohydrate is digested and absorbed as glucose, which is then used as a major source of cytosolic acetyl-CoA for DNL within the lactating mammary gland [6]. Glucose is converted to cytosolic acetyl-CoA through oxidation to pyruvate by the glycolysis pathway in the cytosol. After transport into the mitochondria, pyruvate can then be converted to acetyl-CoA and condensed with oxaloacetate to generate citrate [7]. In tissues performing large amounts of DNL, citrate is transported out of the mitochondria to the cytosol and converted back into acetyl-CoA and oxaloacetate via the enzyme ATP-citrate lyase (*Acly*) [8]. In addition to glucose oxidation via glycolysis, glucose is also oxidized by the pentose phosphate pathway to generate nicotinamide adenine dinucleotide phosphate (NADPH), which is required for fatty acid and cholesterol synthesis [9]. Therefore, in monogastric species, glucose serves as a critical source of both cytosolic acetyl-CoA and NADPH for DNL within the lactating mammary gland.

In mice, the lactating mammary gland supports the production of approximately 5 milliliters of milk per day of which 30–40% is lipid content [10]. This milk serves as the sole source of nutrients to support newborn pup growth and development. The lipid content of milk is composed of fatty acids, fatty acid-containing triglycerides, and cholesterol, which are required for proper growth and development of nursing pups. Triglycerides make up more than 98% of the lipid content of milk with approximately half derived from DNL and the other half acquired from the uptake of fatty acids from the plasma [11, 12]. The contribution of DNL to the lipid content of milk is stable, as increasing dietary fat intake does not alter the proportion of DNL-derived milk triglycerides [13]. In support of the importance of mammary gland DNL, deletion of fatty acid synthase (*Fasn*) in mammary epithelial cells causes a significant reduction in milk triglycerides and decreases the body weight of nursing pups [14]. Furthermore, deletion of Diacylglycerol o-acyltransferase 1 (*Dgat1*) in mice, which catalyzes the formation of diacylglycerol to triacylglycerol results in the complete absence of milk production [15]. Milk cholesterol levels, while relatively low compared to triglyceride concentrations, are stable as modulating dietary cholesterol or hypercholesterolemic mouse models does not affect milk cholesterol content [16]. Taken together, it is clear that mammary gland DNL is critical for milk lipid production and nursing pup growth.

Transcriptomic analyses have defined changes in mammary gland gene expression that occur during lactation. In particular, genes involved in glucose uptake (*Slc2a1*), glycolysis (*Aldoc*, *Pgam1*, *Pkm2*), pentose phosphate pathway (*G6pd2*, *Pgd*, *Rpia*, *Tkt*), tricarboxylic acid (TCA) cycle (*Pc*, *Cs*, *Dbt*), mitochondrial citrate shuttle (*Slc25a1*), fatty acid synthesis (*Acly*, *Fasn*, *Scd1*, *Scd2*, *Fads1*, *Elovl1*), and cholesterol synthesis (*Fdps*, *Sqle*, *Lss*, *Dhcr7*, *Mvd*, *Idi1*, *Fdft1*) are all highly upregulated during lactation [1, 17]. Collectively, these upregulated genes function to allow glucose to be absorbed by the mammary gland and processed through glycolysis, the pentose phosphate pathway, and the TCA cycle to ultimately supply cytosolic acetyl-CoA and NADPH required for DNL of fatty acids and cholesterol. Among these genes, the glycolytic enzyme, Aldolase C (*Aldoc*), is the most upregulated gene in the glycolytic pathway and is highly expressed within the lactating mammary gland [1]. Additionally, *Aldoc* is preferentially expressed within alveolar buds of the mammary gland, the site of milk synthesis [18]. This data, coupled with the recent identification of *Aldoc* as a significant regulator of plasma lipids and liver glucose flux to *de novo* cholesterol biosynthesis *in vitro*, suggest *Aldoc* may be a contributing factor to mammary gland DNL [19].

*Aldoc* is one of three aldolase isozymes, in addition to aldolase A (*Aldoa*) and aldolase B (*Aldob*), that perform the reversible cleavage of fructose 1,6-bisphosphate (F1,6-bisP) into dihydroxyacetone phosphate (DHAP) and glyceraldehyde-3-phosphate (GA3P) in glycolysis.

The physiological role of *Aldoc* in lactation has not been previously investigated; however, multiple lines of evidence would suggest that *Aldoc* is involved in this process. *Aldoc* is transcriptionally regulated within the lactating mammary gland, highly expressed within the mammary alveolar bud, and is a transcriptional target of sterol regulatory element binding proteins 1 and 2 (Srebp1 and Srebp2) [1, 18, 20]. Recently, within the liver, *Aldoc* was shown to contribute to DNL as well as cholesterol and triglyceride metabolism [19]. We therefore tested the hypothesis that *Aldoc* plays a physiological role in lactation by promoting DNL and supporting milk lipid content. Our findings provide functional evidence for the role of *Aldoc* in lactation and indicate *Aldoc* is necessary for normal pup growth and regulates carbohydrate and lipid composition of milk.

## Materials and methods

### Animals

Whole body knockout mice for *Aldoc* were described here [19]. Experimental mice were obtained from heterozygous Aldoc$^{+/-}$ mouse crosses to obtain Aldoc$^{+/+}$ and Aldoc$^{-/-}$ dams. Initial investigation of the transcriptional regulation of the aldolase isozymes was performed in n = 4 mice of each genotype and lactation point (virgin or lactation day 14). The first functional evidence for Aldoc in lactation came from homozygous crosses and analysis of pup weight at weaning. To follow-up these initial studies, we performed longitudinal lactation studies on Aldoc$^{+/+}$ (n = 9) and Aldoc$^{-/-}$ (n = 7) dams at lactation day 3 (L3) and 10 (L10) and euthanized mice at lactation day 14 (L14). For each dam, we measured carbohydrate and lipid content from milk collected at L3 and L10. For a single Aldoc$^{-/-}$ mouse sample the L10 milk collection day fell on Thanksgiving and no milk was collected. This sample was excluded from the lipidomics analysis. Additional mice were used to get histology samples of L10 mammary gland. Mice were ad lib fed a standard rodent chow diet (Envigo 2020x). At the end of animal experiments, mice were euthanized via isoflurane inhalation followed by cervical dislocation. All experimental procedures were performed with approval from the Institutional Care and Use Committee (IACUC) at the University of Wisconsin-Madison.

### Milk collection

Milking was performed at L3 and L10. These time points were chosen to collect milk in order to perform a detailed assessment of milk composition at an early time point (L3) and a midpoint (L10). This allowed us to monitor changes in milk composition over time as well as assess changes in milk composition between these time points. All milk collection was conducted within a four-month timeframe and was performed by removing pups from the dam and placing them in a well ventilated, secure and clean box (inside of the cage) with used bedding from within the cage. After 3.5 hours of separation, dams were anesthetized under constant flow of isoflurane and oxygen, subcutaneously injected with 4 USP units of oxytocin (Bimeda), and mammary glands were manually stimulated while milk was collected into capillary tubes.

### Histology

The inguinal mammary gland was excised from Aldoc$^{+/+}$ and Aldoc$^{-/-}$ dams at lactation day 10 and fixed overnight in 10% buffered formalin, and processed for embedding in paraffin. Paraffin embedded sections were cut on microtome onto microscope slides and stained with hematoxylin and eosin (H&E). Representative images of H&E stained were captured with Life Technologies EVOS FL Auto microscope and software at 10x magnification.

## Lactose, galactose, protein, total cholesterol, triglycerides and glucose analysis

Milk was diluted 1:20 in molecular biology grade water before analysis. Lactose and galactose (assay run without lactase) concentrations were determined using a lactose assay kit (Sigma-Aldrich, MAK017). Protein content was quantified using a Pierce BCA protein assay kit (Thermo Scientific, 23225). Milk, plasma and liver total cholesterol (C7510), triglycerides (T7532) and glucose (G7521) were analyzed with indicated colorimetric assays (Pointe Scientific). Liver lipids were extracted using the Folch method and analyzed using above colorimetric assays [21].

## Untargeted lipidomics

Lipids were extracted from milk after diluting milk samples 1:10 in LC/MS grade water. 5 µl of diluted milk was used in the extraction. Extraction was performed as described here [22]. Organic phase was dried down and resuspended in 240 µl of 65:30:5 (ACN:IPA:$H_2O$). Untargeted lipidomics was performed using a thermo Q Exactive instrument coupled to a vanquish UHPLC. The data were collected in two separate runs in negative or positive mode using tandem mass spectrometry with the top 5 most abundant peaks having ms/ms data collected. Lipids were separated on a 2.1 x 100 mm, 1.7 µM Acquity UPLC CSH C18 Column. The solvents used were A: 70:30 ACN:$H_2O$ (v:v), 10 mM ammonium acetate, 0.025% acetic acid and B: 90:10 IPA:ACN (v:v), 10 mM ammonium acetate, 0.025% acetic acid. The gradient was 0 min, 98% A; 2 min, 98% A; 5 min, 70%A; 19 min, 85% A; 20 min, 1%A; 27 min, 1%A; 28 min 98% A; 32 min 98% A. The flow rate was 0.4 ml/min and the column temperature was 50˚C. Data was analyzed using Lipidex [23]. MS1 data spectra were analyzed using MZmine 2 [24]. Identified triacylglycerol species from each timepoint were normalized to the median peak area of all detected peaks identified using MZmine 2.

## Size-exclusion chromatography of mouse plasma

Size-exclusion chromatography was performed on an AKTA FPLC (Amersham pharmacia biotech). Equivalent volumes of plasma from each group of mice were pooled, totaling 500 µl. Plasma was diluted 1:1 in PBS and was applied to a Superose 6 (GE Pharmacia 17-0537-01) followed in tandem by a Superdex 200 (GE Pharmacia 17-1088-01) column and separated into lipoprotein classes in 10mM PBS, pH 7.4, containing 0.02% sodium azide and collected into 48, 0.5 ml fractions. Fractions were then analyzed for cholesterol (Point Scientific C7510).

## Quantitative PCR

Total RNA was extracted in Qiazol reagent (Qiagen, 79306) according to manufacturers' recommendations. 1000 ng of total RNA was reversed transcribed to cDNA by High-Capacity DNA Reverse Transcription Kit (Thermo Fisher Scientific, 4368813). The qPCR assay was performed using KAPA-SYBR-FAST qPCR master mix kit (Roche, KK4611) in a Roche Lightcycler 480 real time PCR machine. The concentration of mRNA targets for each sample were calculated by the Roche Lightcycler 480 software based off a standard curve and each target mRNA was normalized to the reference gene, Rpl4. Primer sequences as follows organized as gene

(Species, forward primer, reverse primer):
Rpl4 (Mouse, AGCAGCCGGGTAGAGAGG, ATGACTCTCCCTTTTCGGAGT),
Aldoa (Mouse, TGGGAAGAAGGAGAACCTGA, GACAAGCGAGGCTGTTGG,
Aldob (Mouse, GGCTGGTCCCTATTGTTGAG, TAGACAGCAGCCAGGACCTT),

Aldoc (Mouse, `CCTGGAGAGGACAAAGGGATA`, `TGCAAGCCCGTTCATCTC`),
Hmgcs1 (Mouse, `TCCCCTTTGGCTCTTTCACC`, `GGGCAACGATTCCCACATCT`),
Hmgcr (Mouse, `CGTGAGGGTCGTCCAATTT`, `TGAACAAGGACCAAGCCTAAA`),
Mvk (Mouse, `GGAGCAACTGGAGAAGCTAAA`, `TGCCAGGTACAGGTAGAGAA`),
Acss2 (Mouse, `CACCTTCTGGCAAACAGAAAC`, `CTACACCGAAGAATGGGAAAGA`),
Acc1 (Mouse, `CCTCCGTCAGCTCAGATACA`, `TTTACTAGGTGCAAGCCAGACA`).

## Statistical analysis

Statistical analysis of genotype effects was performed using student's unpaired two-tailed t tests. Statistical significance was defined as follows. $*p < 0.05$, $**p < 0.01$ and $***p < 0.001$. Data presented as mean ± standard deviation with individual values. Pearson correlation coefficients were calculated to determine correlations between litter size and Dam traits. Multiple t-tests (no correction for multiple testing) were used to determine significantly altered ($p < 0.05$) TG species identified by untargeted lipidomics. Statistical analysis was performed using GraphPad prism.

## Results

### Mammary *Aldoc* is transcriptionally regulated during lactation

We determined the mammary expression of *Aldoc* in virgin mice and lactating dams at lactation day 14 (L14) of Aldoc$^{+/+}$ and Aldoc$^{-/-}$ mice. Compared to virgin mice, we identified that *Aldoc* expression was significantly increased 20-fold in Aldoc$^{+/+}$ dams at L14 and there was no detectable expression of *Aldoc* in Aldoc$^{-/-}$ dams (Fig 1A). Expression of the related aldolase isozymes, *Aldoa* and *Aldob*, revealed a significant decrease in the expression of both aldolases within the mammary gland at L14 compared to virgin mice in both Aldoc$^{+/+}$ and Aldoc$^{-/-}$ mice (Fig 1A). We also analyzed the expression of *Aldoc*, *Aldoa*, and *Aldob* within the liver of virgin mice and L14 dams and identified that *Aldoc* expression is also significantly increased within the liver of L14 dams and there was no detectable expression of *Aldoc* in Aldoc$^{-/-}$ mice (Fig 1B). *Aldoa* expression was significantly decreased within the liver of L14 mice compared to virgin mice and *Aldob* expression was unchanged (Fig 1B). These data demonstrate that *Aldoc* is the only aldolase isozyme transcriptionally upregulated within the mammary gland as well as the liver during lactation, which is consistent with a previous report that identified *Aldoc* as a gene upregulated in the mammary gland during lactation [1]. These data also demonstrate that Aldoc$^{-/-}$ mice do not express any detectable transcript for the *Aldoc* gene within the liver or mammary gland.

### *Aldoc* deficiency reduces pup weight during lactation

To determine the physiological consequence of *Aldoc* during lactation, we bred wildtype (Aldoc$^{+/+}$) mice and homozygous knockout (Aldoc$^{-/-}$) mice separately and measured the body weight of pups at weaning. Compared to pups from Aldoc$^{+/+}$ dams, pups from Aldoc$^{-/-}$ dams had a significantly reduced body weight (Fig 1C). To focus on the role of maternal *Aldoc* expression during lactation, we used a breeding strategy where all suckling pups would be heterozygous (Aldoc$^{+/-}$) for *Aldoc*, by mating wildtype (Aldoc$^{+/+}$) female dams with homozygous knockout (Aldoc$^{-/-}$) male sires and homozygous knockout (Aldoc$^{-/-}$) female dams with wildtype (Aldoc$^{+/+}$) female dams (Fig 1D). From this strategy, Aldoc$^{+/-}$ pups nursing to Aldoc$^{-/-}$ dams had a significantly reduced body weight at L3, L10, and L14, compared to Aldoc$^{+/-}$ mice nursing to Aldoc$^{+/+}$ dams (Fig 1E). These data demonstrate that pups nursing to Aldoc$^{-/-}$ dams

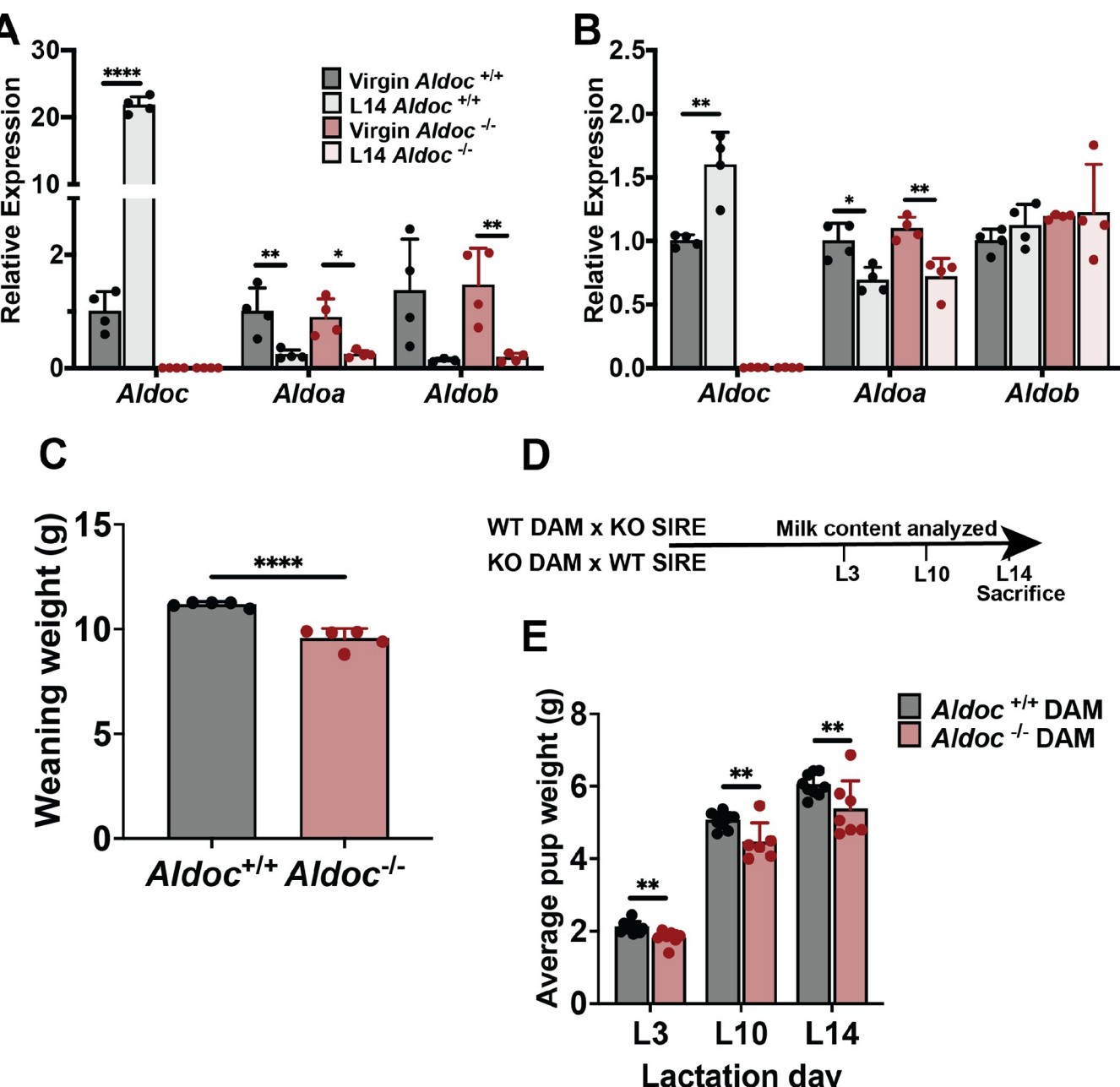

**Fig 1. Mammary *Aldoc* is strongly upregulated during lactation.** Relative expression of the three aldolases, *Aldoc*, *Aldoa*, *Aldob* in lactating day 14 (L14) or virgin (A) mammary gland and (B) liver of Aldoc^+/+ and Aldoc^-/- mice. (C) Body weight of pups at weaning from homozygous crosses for Aldoc^+/+ and Aldoc^-/- mice. (D) Breeding strategy and experimental scheme to generate heterozyous Aldoc^+/- pups suckling to Aldoc^+/+ (WT) and Aldoc^-/- (KO) dams. (E) Pup weight of Aldoc^+/- mice born to Aldoc^+/+ and Aldoc^-/- dams at L3, L10, and L14. Data presented as mean ± SD. Statistical differences determined by unpaired two-tailed t test denoted by *P < 0.05, **P < 0.01 and ****P < 0.0001.

have significantly reduced body weight during lactation independent of the genotype of the pup and suggest maternal *Aldoc* deficiency impacts lactation.

Aldoc deficient mice had similar total body weight, litter size, liver weight, and mammary weight when compared to wild-type mice via a student's two-tailed t test (Fig 2A–2C). Independent of the genotype of the dam (Aldoc^+/+ or Aldoc^-/-), there was a significant correlation between the number of pups in a litter and the body weight of the dam (Fig 2D). Additionally,

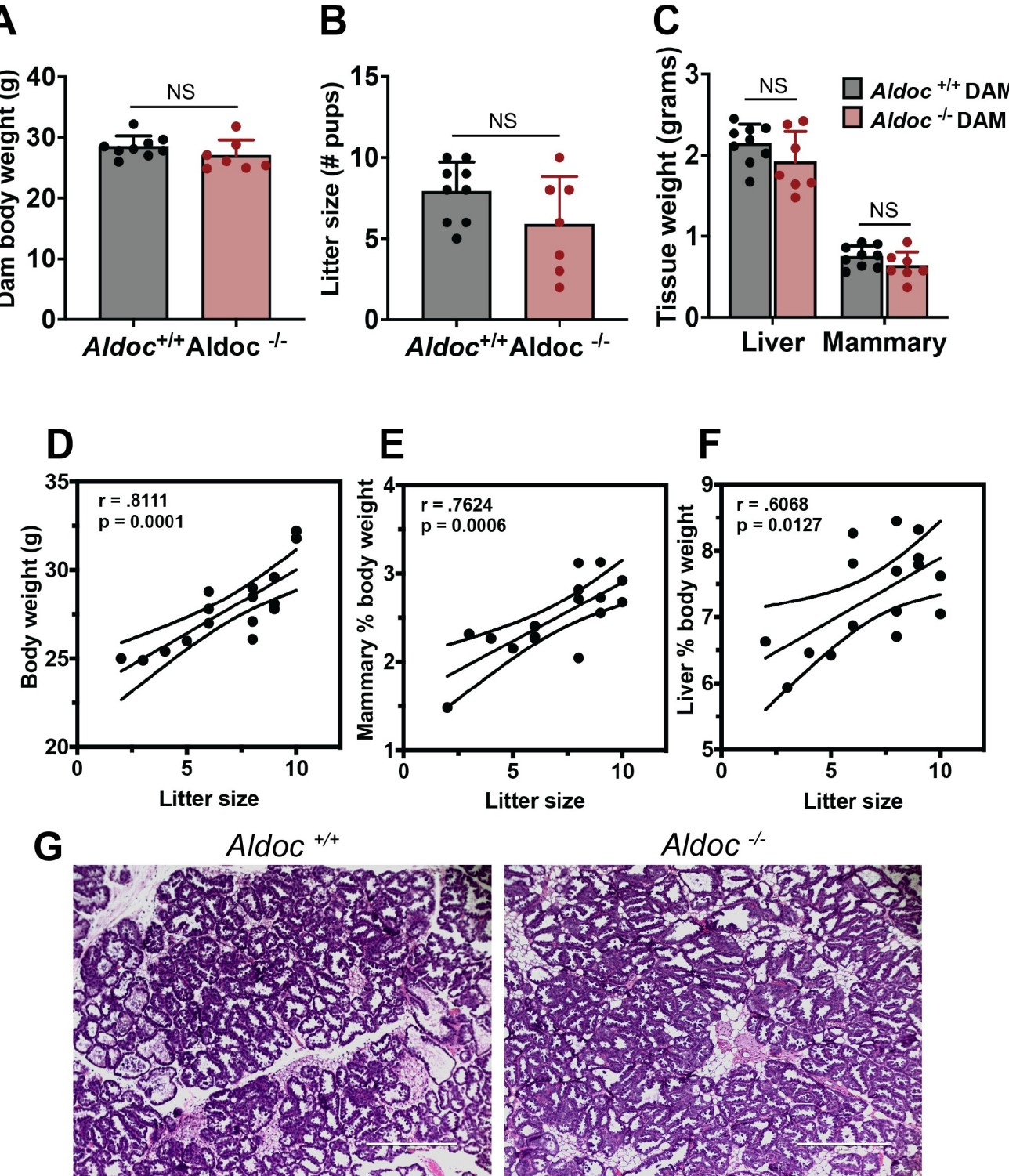

**Fig 2.** ***Aldoc* does not influence dam litter size or weight.** (A) Body weight of Aldoc[+/+] and Aldoc[-/-] dams at L14. (B) Litter size (number of pups) born to Aldoc[+/+] and Aldoc[-/-] dams. (C) Tissue weight of the liver and inguinal mammary gland at L14. Pearson correlation between litter size and (D) body weight of dam, (E) Mammary gland weight as percent of body weight, and (F) liver weight as a percent of body weight with indicated correlation (r) value and p-value (p). (G) Inguinal mammary gland from Aldoc[+/+] and Aldoc[-/-] dams at L10 stained with hematoxylin & eosin. Data presented as mean ± SD. Statistical differences determined by unpaired two-tailed t test, NS = not significant.

there was a significant correlation between the number of pups in a litter and the weight of the inguinal mammary gland and liver, as a percentage of body weight (Fig 2E and 2F). To rule out the possibility that *Aldoc* deficiency might influence mammary gland development during lactation, we also collected the inguinal mammary gland at L10 and observed no histological differences in mammary glands between Aldoc$^{+/+}$ and Aldoc$^{-/-}$ dams (Fig 2G). Both genotypes had abundant ductal-alveolar structures with enlarged milk-producing alveoli. Collectively, besides reduced pup weight, *Aldoc* deficiency did not impact body weight, litter size, liver weight, or inguinal mammary gland weight and structure.

### *Aldoc* contributes to regulating milk composition during lactation

The growth of pups during lactation is supported by essential nutrient acquisition from milk. Our data indicates *Aldoc* does not impact any parameters of lactation besides pup body weight, therefore, we assessed milk composition collected from Aldoc$^{+/+}$ and Aldoc$^{-/-}$ lactating dams at L3 and L10. Saccharide analysis of the milk revealed glucose concentration was similar between Aldoc$^{-/-}$ and Aldoc$^{+/+}$ dams with an increase ($p = 0.054$) in milk glucose concentration at L3 in the Aldoc$^{-/-}$ dams (Fig 3A). Additionally, milk glucose concentration was stable from L3 to L10 (Fig 3B). Milk galactose concentration was increased in Aldoc$^{-/-}$ milk at L3 and L10, which increased between the two timepoints (Fig 3C and 3D). The disaccharide of glucose and galactose, lactose, was unchanged between genotypes at L3, however, there was a significant decrease in lactose concentration at L10 in Aldoc$^{-/-}$ dams compared to Aldoc$^{+/+}$ dams (Fig 3E). Milk lactose concentration in both genotypes increased from L3 to L10, however milk lactose content increased significantly less in in Aldoc$^{-/-}$ milk compared to Aldoc$^{+/+}$ milk (Fig 3F). These data indicate that *Aldoc* deficiency results in an altered saccharide milk profile characterized by increased galactose concentration and decreased lactose, suggesting impaired lactogenesis.

We hypothesized that *Aldoc* would have a physiological role in mammary gland DNL. Triglyceride (glycerol) analysis of milk revealed unaltered triglyceride concentration at L3 and a trend towards lower concentration ($p = 0.12$) at L10 in Aldoc$^{-/-}$ milk compared to Aldoc$^{+/+}$ milk (Fig 4A). However, milk triglyceride concentration decreased from L3 to L10 in the Aldoc$^{-/-}$ dams and remained stable in Aldoc$^{+/+}$ dams (Fig 4B). Milk cholesterol concentration was assessed and revealed similar cholesterol concentration at L3, while there was a significant reduction in cholesterol concentration at L10 in Aldoc$^{-/-}$ milk (Fig 4C). In Aldoc$^{+/+}$ dams, milk cholesterol concentration increased from L3 to L10, while cholesterol concentration was unchanged from L3 to L10 in Aldoc$^{-/-}$ dams (Fig 4D). Finally, we measured milk protein concentration and found no difference in protein concentration at L3 or L10 between Aldoc$^{-/-}$ and Aldoc$^{+/+}$ dams (Fig 4E). Milk protein concentration increased from L3 to L10 in both Aldoc$^{+/+}$ and Aldoc$^{-/-}$ dams (Fig 4F). Our analysis of milk lipid concentration identified a significant decrease in triglyceride concentration from L3 to L10 during lactation in Aldoc$^{-/-}$ dams and significantly lower milk cholesterol concentration at L10. Collectively, these data suggest impaired DNL of triglycerides and cholesterol within the lactating mammary gland of Aldoc$^{-/-}$ dams.

### *Aldoc* alters milk medium and long-chain fatty-acid containing triglycerides

In previous work, we demonstrated that *Aldoc* plays a role in contributing to DNL within the liver [19]. Based on this work and the hypothesis that *Aldoc* would play a role in mammary gland DNL, we performed untargeted lipidomics to quantify milk triglycerides at L3 and L10. Milk lipidomics revealed an increase in triglycerides containing medium chain fatty acids (TG

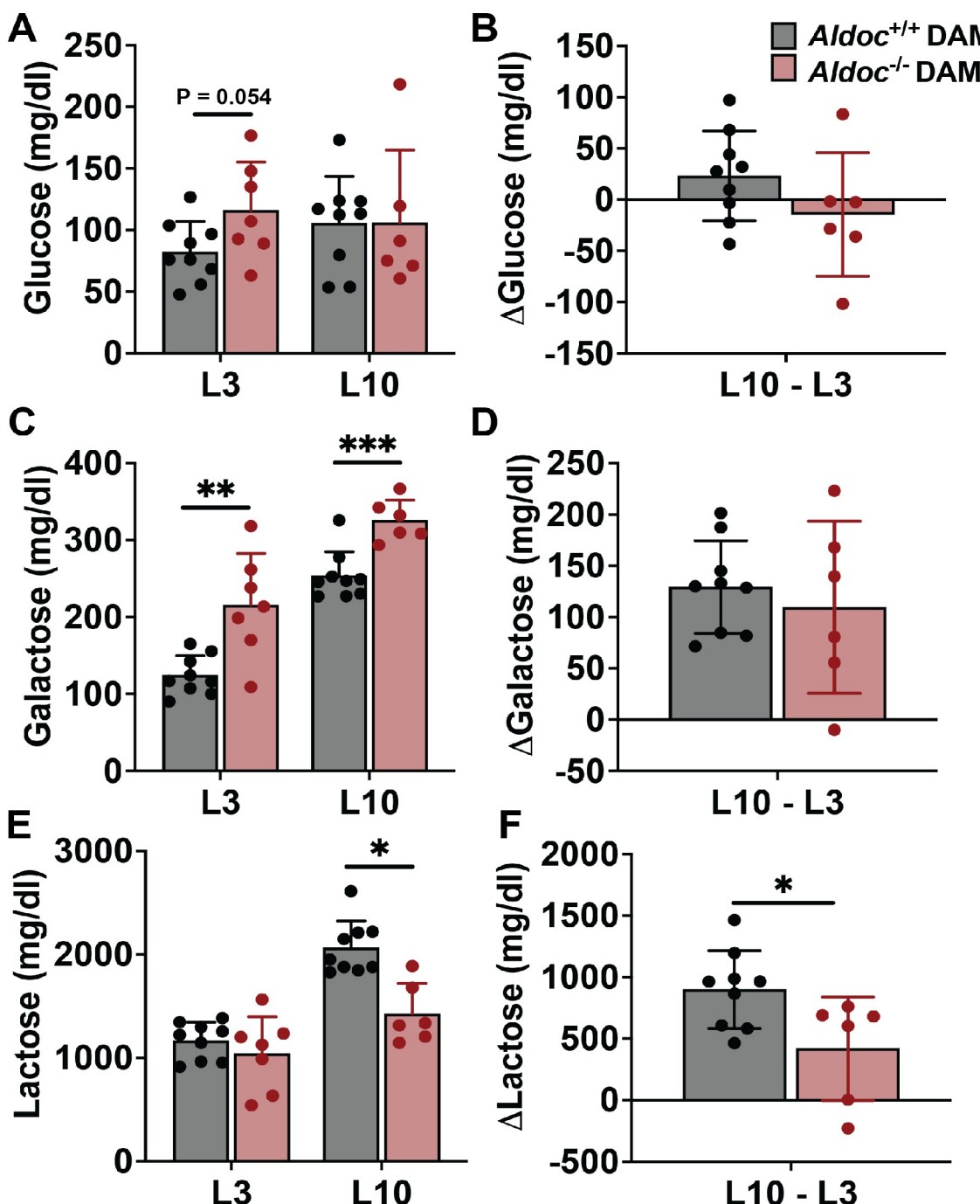

**Fig 3. Aldoc$^{-/-}$ dams are unable to maintain milk lactose content during lactation.** (A) Milk glucose concentration at L3 and L10. (B) Change in milk glucose concentration from L3 to L10. (C) Milk galactose concentration at L3 and L10. (D) Change in milk galactose concentration from L3 to L10. (E) Milk lactose concentration at L3 and L10. (F) Change in milk lactose concentration from L3 to L10. Data presented as mean ± SD. Statistical differences determined by unpaired two-tailed t test denoted by *P < 0.05, **P < 0.01 and ***P <0.001.

36:0 to TG 44:2) from L3 to L10 (Fig 5A). We also identified 20 unique triglycerides that were significantly lower in Aldoc$^{-/-}$ milk at L10, many of which were the same triglycerides that increased from L3 to L10 in Aldoc$^{+/+}$ milk (Fig 5B). Overall, this data demonstrates that

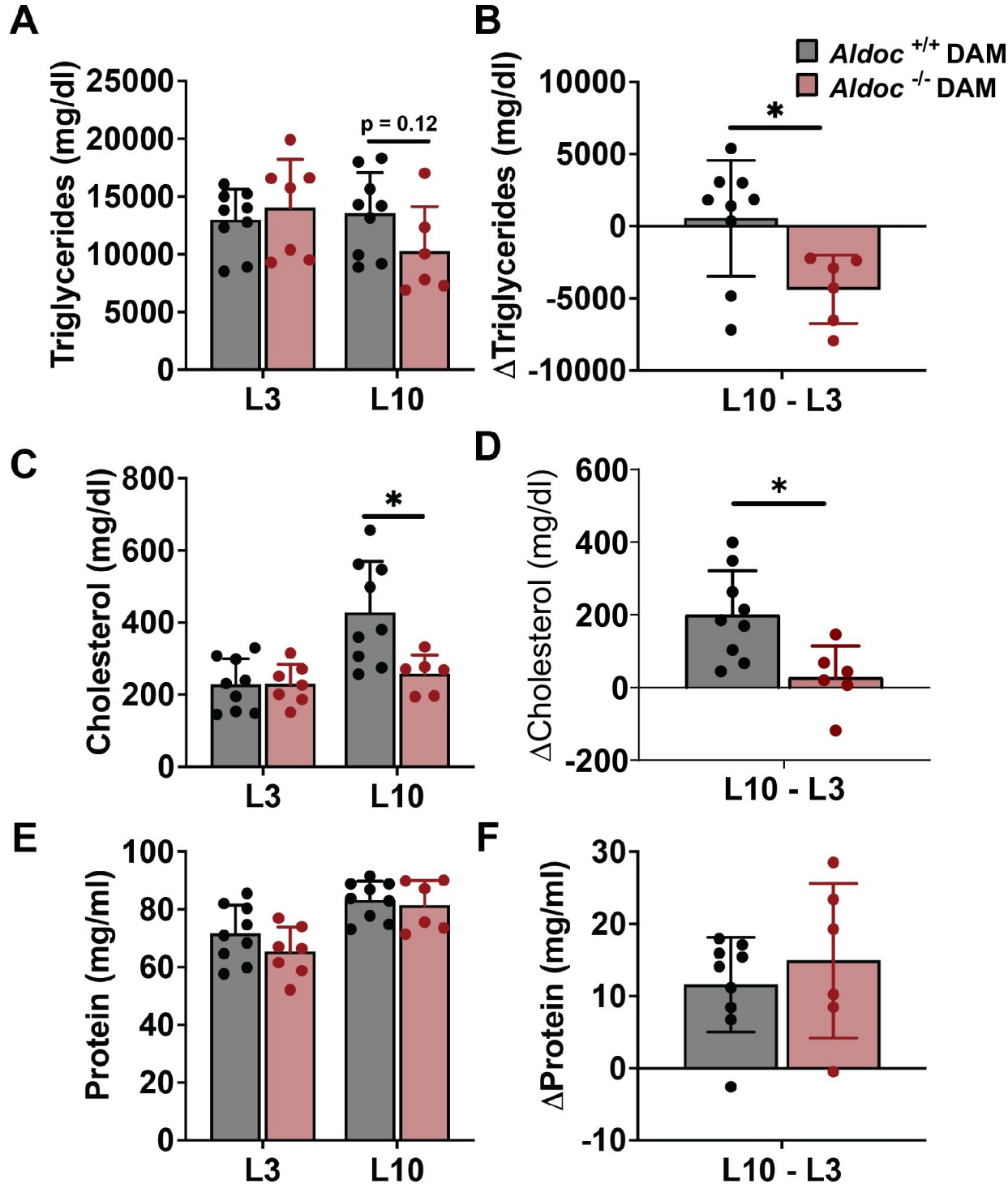

**Fig 4. Aldoc⁻/⁻ dams are unable to maintain milk lipid content during lactation.** (A) Milk triglyceride concentration at L3 and L10. (B) Change in milk triglyceride concentration from L3 to L10. (C) Milk cholesterol concentration at L3 and L10. (D) Change in milk cholesterol concentration from L3 to L10. (E) Milk protein concentration at L3 and L10. (F) Change in milk protein concentration from L3 to L10. Data presented as mean ± SD. Statistical differences determined by unpaired two-tailed t test denoted by *P < 0.05 ** P < 0.01.

Aldoc$^{-/-}$ dams produce milk containing significantly reduced triglycerides with medium to long chain fatty acids, which is suggestive of impaired DNL within the mammary gland of Aldoc$^{-/-}$ dams.

### *Aldoc* deficiency impacts plasma lipids and liver during lactation

The liver is an additional source of de novo synthesized triglyceride and cholesterol which is delivered to other tissues via the bloodstream [25]. Therefore, we analyzed plasma and liver lipid phenotypes in Aldoc$^{+/+}$ and Aldoc$^{-/-}$ dams at L14. There were no significant differences in plasma cholesterol, triglycerides, or glucose in Aldoc$^{+/+}$ or Aldoc$^{-/-}$ dams (Fig 6A–6C). Aldoc$^{-/-}$ dams had a trend towards increased plasma cholesterol (p = 0.11), therefore, we used size exclusion chromatography to separate plasma lipoproteins into low-density lipoprotein (LDL) cholesterol and high-density lipoprotein (HDL) cholesterol. Separation of plasma lipoproteins revealed that Aldoc$^{-/-}$ dams had increased LDL cholesterol compared to Aldoc$^{+/+}$ dams (Fig 6D). Liver lipid isolation and analysis revealed no difference in cholesterol concentration within the livers of Aldoc$^{+/+}$ and Aldoc$^{-/-}$ dams (Fig 7A). However, there was a significant increase in liver triglyceride concentration in Aldoc$^{-/-}$ dams compared to Aldoc$^{+/+}$ dams (Fig 7B). Due to the increased plasma levels of cholesterol and increased liver triglyceride content in Aldoc$^{-/-}$ dams, we measured gene expression of key enzymes in the synthesis of cholesterol and fatty acids. In the cholesterol biosynthetic pathway, we identified that Aldoc$^{-/-}$ dams have significantly higher relative expression of *Hmgcr* (3-Hydroxy-3-Methylglutaryl-CoA Reductase), with no change in *Hmgcs1* (3-Hydroxy-3-Methylglutaryl-CoA Synthase 1) or *Mvk* (Mevalonate Kinase) (Fig 7C). Also, genes involved in the biosynthesis of fatty acids, including *Acss2* (Acyl-CoA Synthetase Short Chain Family Member 2), *Fasn* (Fatty Acid Synthase), and *Acc1* (Acetyl-CoA Carboxylase Alpha) were unchanged between Aldoc$^{+/+}$ and Aldoc$^{-/-}$ dams (Fig 7C). Measuring the liver mRNA expression of the aldolase isozymes, *Aldoa* and *Aldob* identified no significant changes, while *Aldoc* expression within the liver was undectable in Aldoc$^{-/-}$ dams (Fig 7C). Taken together, this data suggests that liver and plasma lipid metabolism does not play a role in the decreased pup growth and altered milk composition seen in Aldoc$^{-/-}$ dams.

## Discussion

In this study we investigated the physiological importance of Aldolase C during lactation. The lactating mammary gland coordinates the production and secretion of milk rich in lipids, carbohydrates, and protein to supply nutrients for growth and development of pups. With regard to milk lipids, the mammary gland strongly induces DNL during lactation to produce triglycerides and cholesterol from glucose [1]. More than 60 years ago, research on mammary gland lactation demonstrated that in monogastric species, like rodents, glucose is processed through glycolysis and the pentose phosphate pathway to supply cytosolic acetyl-CoA and NADPH required for DNL [9]. Using transcriptomics, previous work established that along with the enzymes of the DNL pathway (*Fasn*, *Hmgcr*, etc), enzymes required for the conversion of glucose to cytosolic acetyl-CoA (*Slc2a1*, *Pdc*, *Cs*, *Slc25a1*, and *Acly*) are all upregulated within the lactating mammary gland of mice [1]. This work also identified that *Aldoc* gene expression is strongly induced during lactation [1]. In related work, *Aldoc* was identified as a prolactin-dependent gene that is highly expressed in the mammary gland, with expression highest in the mammary alveolar buds, the site of milk synthesis [18]. Despite being recognized as a highly expressed gene in the lactating mammary gland, no studies have been conducted to determine the physiological role of *Aldoc* in lactation.

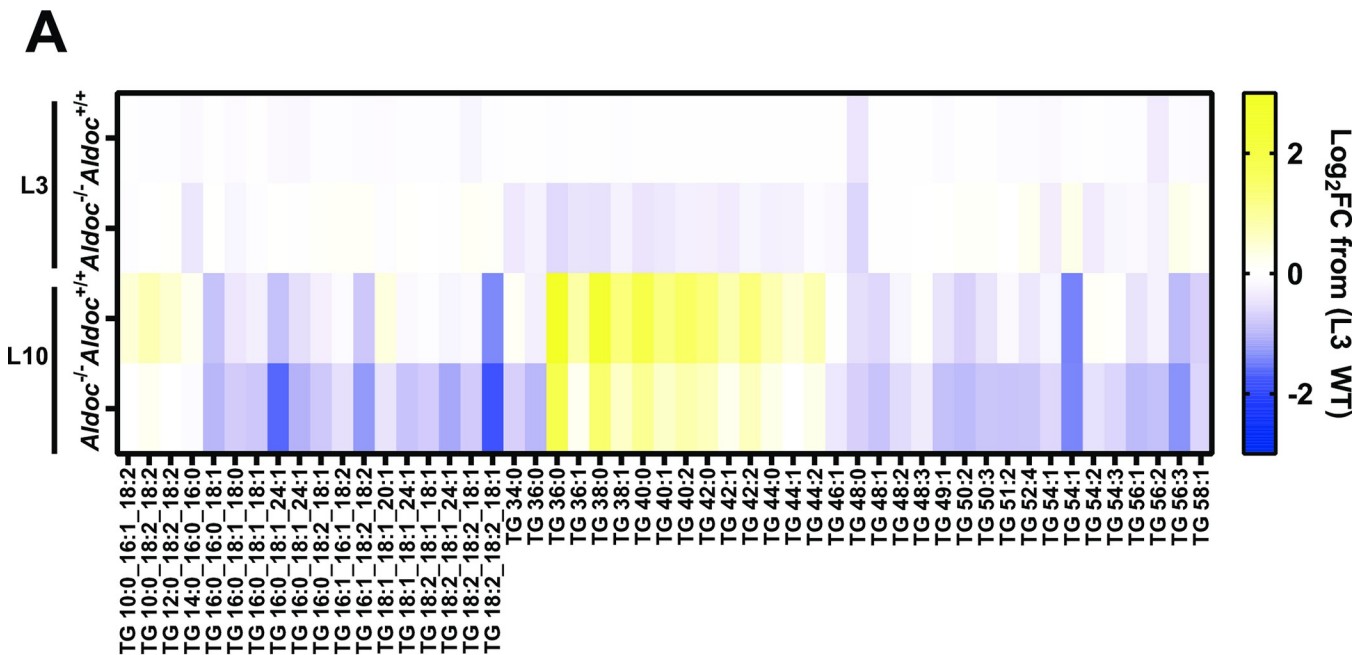

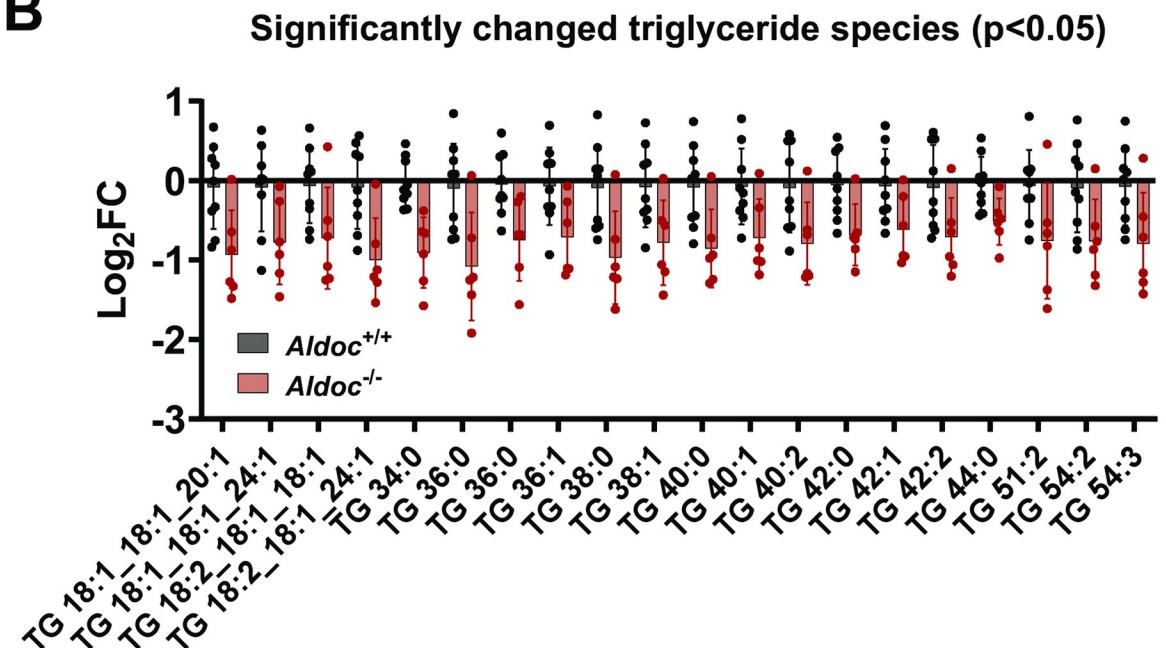

**Fig 5. Aldoc⁻/⁻ dams have reduced triglycerides with medium and long-chain milk fatty acids.** (A) Heatmap of 51 triglyceride species in Aldoc⁺/⁺ (WT) and Aldoc⁻/⁻ (KO) at L3 and L10 showing the Log₂ fold change relative to Aldoc⁺/⁺ (WT) dams at L3 (B) Significantly changed triglyceride species within milk of Aldoc⁻/⁻ dams compared to Aldoc⁺/⁺ dams at L10.

The physiological importance of *Aldoc* was first demonstrated by decreased pup weight at weaning in mice born to Aldoc⁻/⁻ dams regardless of pup genotype. Analysis of milk saccharides revealed similar glucose concentration between genotypes which provides evidence that mammary gland glucose uptake is normal in Aldoc⁻/⁻ dams [26]. However, saccharide analysis also revealed an increase in milk galactose and decrease in lactose in Aldoc⁻/⁻ milk providing

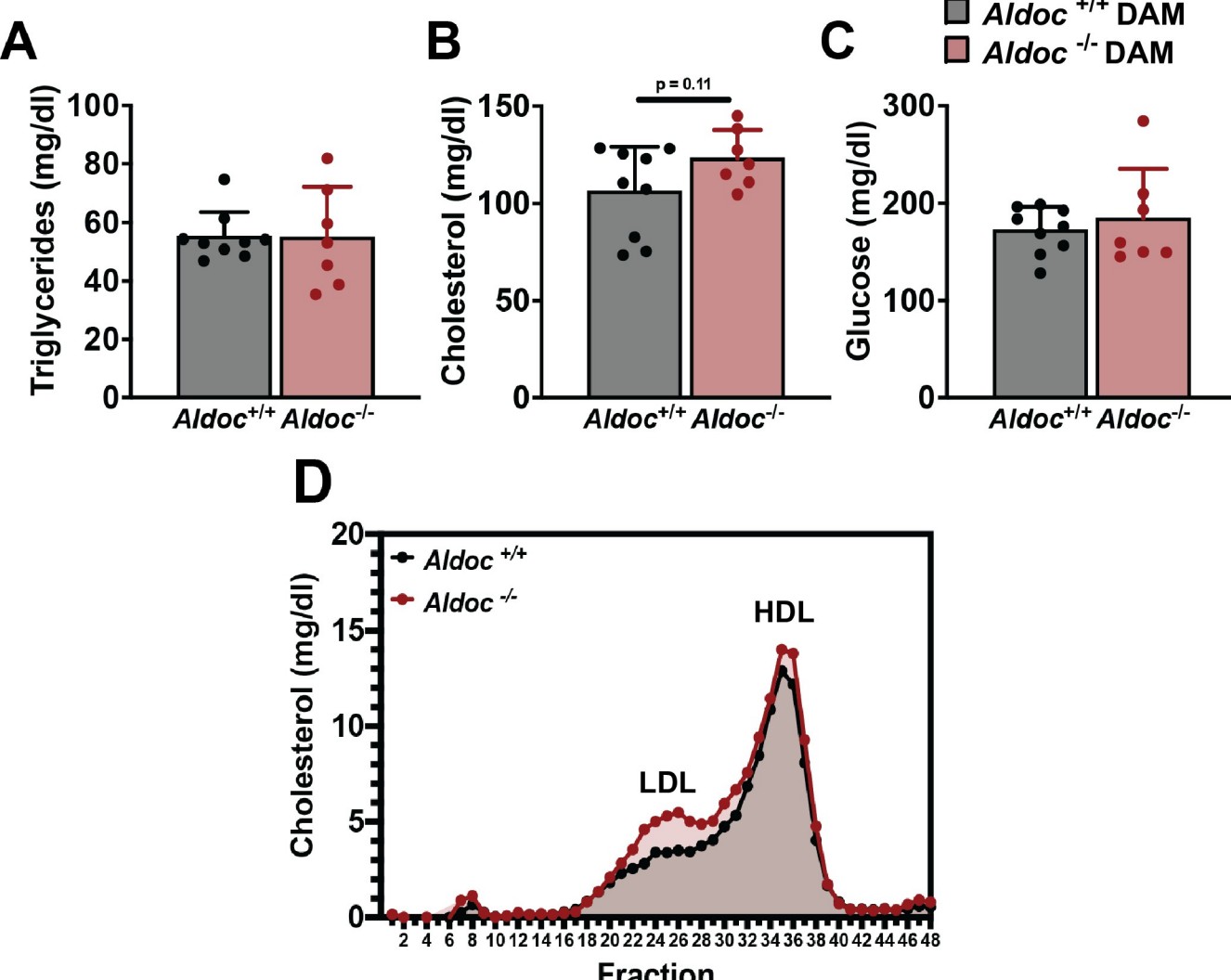

**Fig 6. Increased plasma LDL cholesterol in Aldoc[-/-] dams.** Plasma concentration of total (A) triglycerides, (B) cholesterol, and (C) glucose in L14 Aldoc[+/+] and Aldoc[-/-] dams. (D) Size exclusion chromatography separation of plasma cholesterol from L14 Aldoc[+/+] and Aldoc[-/-] dams to identify LDL cholesterol and HDL cholesterol. Data presented as mean ± SD. Statistical differences determined by unpaired two-tailed t test.

the first evidence that *Aldoc* is physiologically important for mammary lactose biosynthesis. Previous work from our lab identified that Aldolase C is a regulator of liver and plasma triglyceride concentrations, therefore we hypothesized that *Aldoc* would also regulate these pathways in the lactating mammary gland [19]. Colorimetric assays for total cholesterol and triglycerides supported this hypothesis, revealing decreases in both lipids across the lactation timepoints as compared to wild-type mice. Untargeted lipidomics of the milk samples demonstrated that at L10, Aldoc[-/-] dams have significantly reduced levels of milk triglyceride species with medium and long-chain fatty acids shorter than 18 carbons, which largely overlap with triglyceride species that are increased from L3 to L10 in Aldoc[+/+] milk. This observation aligns well with data in humans that shows that the mammary gland has a strong preference for the utilization of glucose in *de novo* synthesis of fatty acids shorter than 16 carbons in length [6, 27]. Milk protein concentration was similar between Aldoc[+/+] and Aldoc[-/-] dams, indicating that pup growth defects are not protein related. Collectively, our study provides the first evidence that

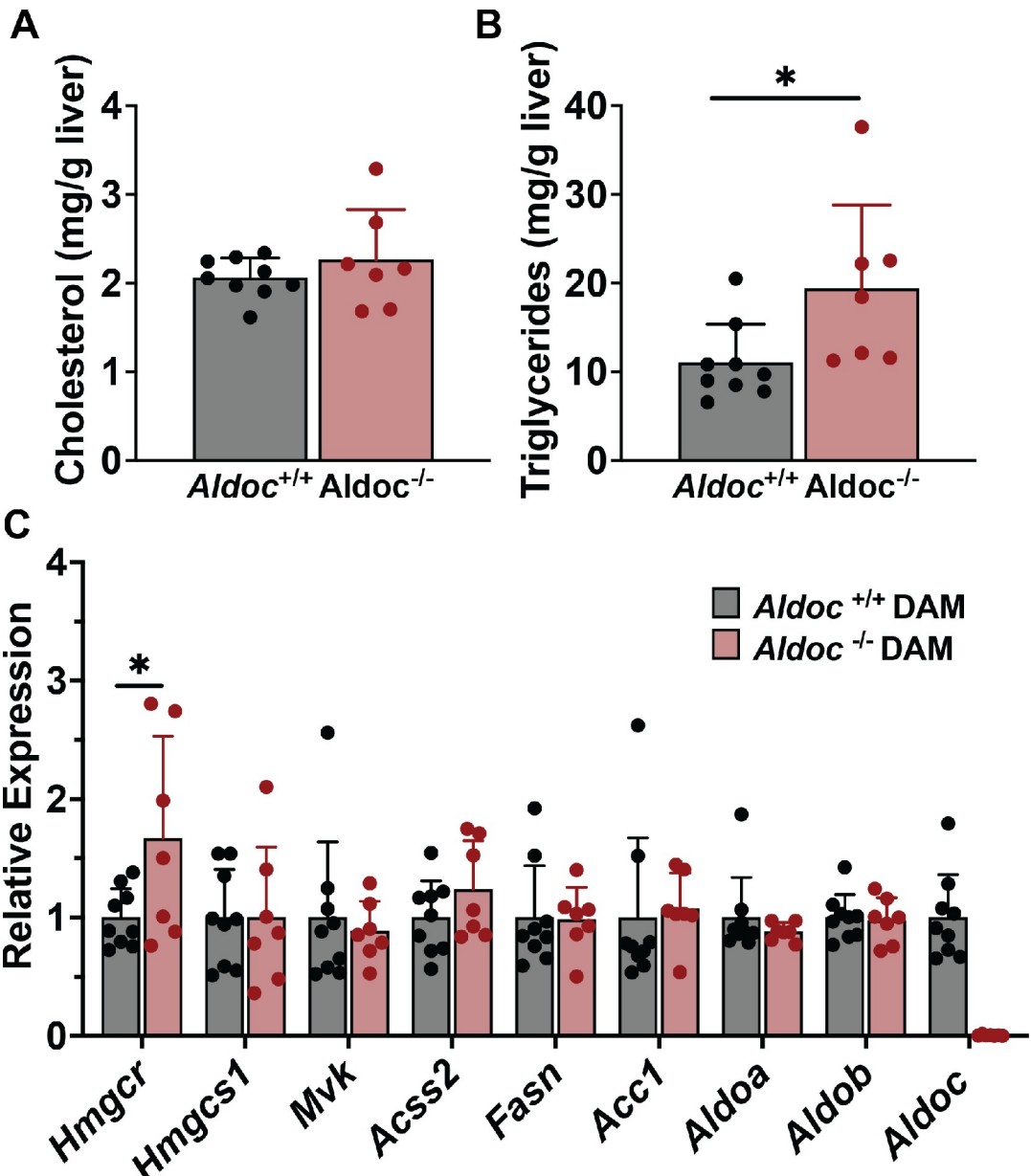

**Fig 7. Increased liver triglycerides in Aldoc^-/- dams.** Liver concentration of (A) cholesterol and (B) triglycerides in Aldoc^+/+ and Aldoc^-/- dams at L14. (C) Liver mRNA expression of essential enzymes in liver cholesterol biosynthesis (*Hmgcr*, *Hmgcs1*, *Mvk*), fatty acid synthesis (*Acss2*, *Fasn*, *Acc1*) and the three aldolase isozymes (*Aldoa*, *Aldob*, *Aldoc*) in Aldoc^+/+ and Aldoc^-/- dams at L14. Data presented as mean ± SD. Statistical differences determined by unpaired two-tailed t test denoted by *P < 0.05.

Aldoc is physiologically important during lactation and regulates milk macronutrient composition. The altered milk composition likely leads to inadequate nutrient acquisition in suckling pups resulting in the observed reduction in body weight in pups suckling to Aldoc^-/- dams.

Our data are consistent with previous studies utilizing genetically engineered mice lacking specific components of the DNL pathway. Deletion of *Fasn*, the required enzyme for fatty acid synthesis, within the mammary epithelial cells, leads to reduced pup body weight and lower milk concentrations of triglycerides [14]. Furthermore, deletion of Srebp cleavage-activating

protein (*Scap*), which is necessary for the transcriptional activity of the Srebp1 and Srebp2 and therefore activation of DNL, led to decreased pup body weight and lower milk concentrations of triglycerides [28]. The strong regulation of *Aldoc* within the mammary gland and the known regulation of *Aldoc* by Srebp1 and Srebp2, strongly suggests that *Aldoc* is a conserved component of the DNL pathways. This idea is additionally supported by our previous studies of *Aldoc*, which demonstrate that *Aldoc* contributes to DNL and influences plasma triglyceride and cholesterol metabolism [19]. Interestingly, *Aldoc* did not negatively regulate liver and plasma triglycerides and total cholesterol in the context of lactation, perhaps suggesting alternative metabolic pathways are responsible for providing acetyl-CoA for liver DNL during lactation.

Here, we tested the physiological role of *Aldoc* during lactation. Our data support the idea that *Aldoc* coordinates the efficient utilization of glucose for not only the generation of cholesterol and triglycerides, but also for lactose synthesis. Mechanistic evidence for how *Aldoc*, a glycolytic enzyme, can regulate the synthesis of lactose, triglycerides, and cholesterol remains unknown. One possible explanation is through regulation of metabolic compartmentalization. In other biological systems, aldolases are known to interact with other glycolytic partners and form a metabolon [29–31]. Strong up-regulation of *Aldoc* within the lactating mammary gland and the milk phenotypes identified within, may indicate that *Aldoc* is important in coordinating this metabolon in the efficient utilization of glucose for both lactose synthesis and generation of glucose-derived acetyl-CoA for DNL. In summary, we have generated the first evidence that *Aldoc* performs a physiological role during lactation to coordinate the production of milk lactose and lipids.

## Conclusions

Despite significant evidence indicating *Aldoc* plays a role in lactation, no study to date has formally tested its role. Here, we performed a series of lactation studies in Aldoc[+/+] and Aldoc[-/-] dams and demonstrate the importance of *Aldoc* in maintaining milk carbohydrate and lipid content during lactation. Based on our studies and data in humans, *Aldoc* may play a role in contributing to human milk carbohydrate and lipid content. Future studies are warranted to understand the biological pathway and molecular mechanisms through which *Aldoc* contributes to DNL and milk production within the mammary gland.

## Supporting information

**S1 Dataset.**
(XLSX)

## Acknowledgments

We would like to acknowledge Hannah Fricke for demonstrating how perform milk collection in mice and Laura Hernandez for her helpful advice.

## Author Contributions

**Conceptualization:** James A. Votava, Brian W. Parks.

**Data curation:** James A. Votava, Jing Fan, Brian W. Parks.

**Formal analysis:** James A. Votava, Brian W. Parks.

**Funding acquisition:** Brian W. Parks.

**Investigation:** James A. Votava.

**Methodology:** James A. Votava, Brian W. Parks.

**Project administration:** James A. Votava, Brian W. Parks.

**Resources:** James A. Votava, Jing Fan.

**Software:** James A. Votava.

**Supervision:** Jing Fan, Brian W. Parks.

**Validation:** James A. Votava.

**Visualization:** James A. Votava.

**Writing – original draft:** James A. Votava.

**Writing – review & editing:** James A. Votava, Jing Fan, Brian W. Parks.

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
