## [Decision Letter · Decision Letter 0]

18 Sep 2024

PONE-D-24-31001Physiological consequences of Aldolase C deficiency during lactationPLOS ONE

Dear Dr. Parks,

Thank you for submitting your manuscript to PLOS ONE. After careful consideration, we feel that it has merit but does not fully meet PLOS ONE’s publication criteria as it currently stands. Therefore, we invite you to submit a revised version of the manuscript that addresses the points raised during the review process.

We look forward to receiving your revised manuscript.

Kind regards,

Mohd Akbar Bhat

Academic Editor

PLOS ONE

“This work was supported in part by NIH R01-HL147097 (B.W.P).”

Reviewers' comments:

Reviewer's Responses to Questions

**Comments to the Author**

1. Is the manuscript technically sound, and do the data support the conclusions?

Reviewer #1: Partly

Reviewer #2: Yes

2. Has the statistical analysis been performed appropriately and rigorously? 

Reviewer #1: No

Reviewer #2: Yes

3. Have the authors made all data underlying the findings in their manuscript fully available?

Reviewer #1: Yes

Reviewer #2: Yes

4. Is the manuscript presented in an intelligible fashion and written in standard English?

Reviewer #1: Yes

Reviewer #2: Yes

5. Review Comments to the Author

Reviewer #1: Overall, the study appears to have been well conducted and provided interesting results, but there are issues to consider.

Introduction:

- The purpose of the introduction is to give a scientific background to the main topic of the study.

- The last paragraph of the introduction should state the rationale and hypothesis of the study.

- Line 113: "Deletion of Aldoc did not impact pregnancy or litter size,……" The introduction is not the appropriate place to present the results of the study!!

Materials and Methods:

- It is very confusing and makes the results and figures difficult to follow and understand.

- It is advisable to provide a representative flowchart of the study protocol.

- Before rushing into the details of biochemical analysis or staining method, you should first clearly define the experimental groups, the number of mice/group and the parameters being measured and in which sample.

- Please specify the dates of milk collection and the biochemical parameters being analyzed.

- Explain the rationale behind choosing L3, L10 or L14, and why not all parameters are analyzed on the same date.

Statistical analysis:

- Did you check for normality of data?

- It is not clear which test was used to compare the means of two different groups, and which test was used to compare the means of the same group at different time points (e.g. L3, L10).

- The two-tailed Student's t-test is used to compare two different groups only. Please check the statistical analysis of your data.

Results:

- Line 197: The word "Aldoa and Aldob" was repeated twice in the same sentence, Please correct.

Reviewer #2: The manuscript presents valuable and novel findings on the physiological role of Aldoc in lactation. It is well-executed, and the results are robust. To improve the clarity of the manuscript, I would recommend authors to address the following comments:

Abstract:

Consider adding a brief sentence on how these findings could impact broader fields, such as human lactation research.

Introduction:

The background on Aldoc's regulatory role in cholesterol and lipid metabolism is relevant. It will benefit from adding a paragraph emphasizing the knowledge gaps about its role in lactation specifically.

Methods:

The methods section is detailed and well-organized, providing sufficient information for replication. The use of multiple techniques (e.g., lipidomics, histology, and gene expression assays) is a strength. However, the section could benefit from a clearer explanation of how some decisions were made regarding time points for analysis (e.g., why L3 and L10 were chosen).

Discussion:

Clarify the distinction between previously published data on Aldoc in other tissues and this study’s findings in the mammary gland and avoid repeating points about metabolic coordination in lactation.

Add a brief conclusion summarizing the implications of Aldoc deficiency in lactation and suggestions for future research directions.

Overall, this manuscript presents significant and original research on the role of Aldoc in lactation. After addressing the minor suggestions above, I recommend that this study be published. The work is a valuable contribution to the field of metabolic regulation during lactation.

6. PLOS authors have the option to publish the peer review history of their article (what does this mean?). If published, this will include your full peer review and any attached files.

Reviewer #1: No

Reviewer #2: No

---

## [Author Response · Author response to Decision Letter 0]

4 Oct 2024

Response to Reviewers

Reviewer #1: Overall, the study appears to have been well conducted and provided interesting results, but there are issues to consider.

Introduction:

- The purpose of the introduction is to give a scientific background to the main topic of the study.

In the manuscript, the introduction provides detailed background on the role of de novo lipogenesis within the mammary gland as well as the transcriptional regulation of this process and known information on aldolase C. 

- The last paragraph of the introduction should state the rationale and hypothesis of the study.

In the revised manuscript, we now provide a concise rationale and hypothesis in the last paragraph of the introduction.

- Line 113: "Deletion of Aldoc did not impact pregnancy or litter size,……" The introduction is not the appropriate place to present the results of the study!!

In the revised manuscript, we have removed specific results and now include an overall statement of findings.

Materials and Methods:

- It is very confusing and makes the results and figures difficult to follow and understand.

In the revised manuscript, we have added relevant information to materials and methods to clarify experimental groups and milk collection time points. The birth dates and milking dates are provided in the supplementary minimal dataset.

- It is advisable to provide a representative flowchart of the study protocol.

A flowchart is included in figure 1D. Additionally, we have added more details to the various groups of mice used in this manuscript.

- Before rushing into the details of biochemical analysis or staining method, you should first clearly define the experimental groups, the number of mice/group and the parameters being measured and in which sample.

In the revised manuscript, we have defined experimental groups, numbers/group, and parameters measured to help clarify. 

- Please specify the dates of milk collection and the biochemical parameters being analyzed.

In the revised manuscript, we have included a statement indicating all milk collection was performed within a four-month timeframe and provided exact dates in the supplementary minimal dataset.

- Explain the rationale behind choosing L3, L10 or L14, and why not all parameters are analyzed on the same date.

In the revised manuscript, we have added details and rationale behind choosing lactation day 3 and 10 for milk collection and analysis. As we collected milk from the same mice a multiple time points (L3 and L10), it is not possible to do this on the same date. However, all biochemical milk composition analyses were conducted on same date. 

Statistical analysis:

- Did you check for normality of data? 

Yes, the data within this manuscript is normally distributed. 

- It is not clear which test was used to compare the means of two different groups, and which test was used to compare the means of the same group at different time points (e.g. L3, L10).

We apologize if this wasn’t clear. In the revised manuscript, each figure legend now indicates the statistical test used and is included in the statistical analysis section of the materials and methods section.

- The two-tailed Student's t-test is used to compare two different groups only. Please check the statistical analysis of your data.

For all of our statistical analyses, we are comparing two groups (Aldoc+/+ vs Aldoc-/-), unpaired two tailed students t-test is the appropriate test for these analyses.

Results:

- Line 197: The word "Aldoa and Aldob" was repeated twice in the same sentence, Please correct.

This is now corrected.

Reviewer #2: The manuscript presents valuable and novel findings on the physiological role of Aldoc in lactation. It is well-executed, and the results are robust. To improve the clarity of the manuscript, I would recommend authors to address the following comments:

Abstract:

Consider adding a brief sentence on how these findings could impact broader fields, such as human lactation research.

We now include a sentence on the impact on human lactation within the abstract.

Introduction:

The background on Aldoc's regulatory role in cholesterol and lipid metabolism is relevant. It will benefit from adding a paragraph emphasizing the knowledge gaps about its role in lactation specifically.

In the revised manuscript, we have modified the last paragraph of the introduction to clarify the knowledge gaps, rationale for study, and hypothesis we are testing.

Methods:

The methods section is detailed and well-organized, providing sufficient information for replication. The use of multiple techniques (e.g., lipidomics, histology, and gene expression assays) is a strength. However, the section could benefit from a clearer explanation of how some decisions were made regarding time points for analysis (e.g., why L3 and L10 were chosen).

In the revised manuscript, we now include rational for milk collection at lactation day 3 and 10. 

Discussion:

Clarify the distinction between previously published data on Aldoc in other tissues and this study’s findings in the mammary gland and avoid repeating points about metabolic coordination in lactation.

 In the revised manuscript, repeated points about metabolic coordination have been removed and previous findings of the role of Aldoc in regulating plasma cholesterol and triglycerides have been clarified. 

Add a brief conclusion summarizing the implications of Aldoc deficiency in lactation and suggestions for future research directions.

 In the revised manuscript, we have added a conclusions section.

Overall, this manuscript presents significant and original research on the role of Aldoc in lactation. After addressing the minor suggestions above, I recommend that this study be published. The work is a valuable contribution to the field of metabolic regulation during lactation.

---

## [Decision Letter · Decision Letter 1]

1 Dec 2024

Physiological consequences of Aldolase C deficiency during lactation

PONE-D-24-31001R1

Dear Dr. Parks

We’re pleased to inform you that your manuscript has been judged scientifically suitable for publication and will be formally accepted for publication once it meets all outstanding technical requirements.

Kind regards,

Mohd Akbar Bhat, Ph.D.

Academic Editor

PLOS ONE

Additional Editor Comments (optional):

The authors have revised this manuscript and incorporated all the recommended suggestions; therefore, I recommend this paper for publication.

Reviewers' comments:

Reviewer's Responses to Questions

**Comments to the Author**

1. If the authors have adequately addressed your comments raised in a previous round of review and you feel that this manuscript is now acceptable for publication, you may indicate that here to bypass the “Comments to the Author” section, enter your conflict of interest statement in the “Confidential to Editor” section, and submit your "Accept" recommendation.

Reviewer #3: (No Response)

Reviewer #4: All comments have been addressed

2. Is the manuscript technically sound, and do the data support the conclusions?

Reviewer #3: Yes

Reviewer #4: Yes

3. Has the statistical analysis been performed appropriately and rigorously? 

Reviewer #3: Yes

Reviewer #4: Yes

4. Have the authors made all data underlying the findings in their manuscript fully available?

Reviewer #3: Yes

Reviewer #4: Yes

5. Is the manuscript presented in an intelligible fashion and written in standard English?

Reviewer #3: Yes

Reviewer #4: Yes

6. Review Comments to the Author

Reviewer #3: the authors had addressed all required revisions no more comments are needed to be fulfil the article is much been improved

Reviewer #4: The authors have addressed all the suggestions raised by the reviewers. Hence, the manuscript can be accepted for the publication.

7. PLOS authors have the option to publish the peer review history of their article (what does this mean?). If published, this will include your full peer review and any attached files.

Reviewer #3: **Yes: **amany A. Saleh

Reviewer #4: **Yes: **Shiwali Goyal

---

## [Editor Report · Acceptance letter]

3 Dec 2024

PONE-D-24-31001R1 

PLOS ONE

Dear Dr. Parks, 

I'm pleased to inform you that your manuscript has been deemed suitable for publication in PLOS ONE. Congratulations! Your manuscript is now being handed over to our production team.

Kind regards, 

on behalf of

Dr. Mohd Akbar Bhat 

Academic Editor

PLOS ONE